# Increased Red Blood Cell Distribution Width in the First Year after Diagnosis Predicts Worsening of Systemic Sclerosis-Associated Interstitial Lung Disease at 5 Years: A Pilot Study

**DOI:** 10.3390/diagnostics11122274

**Published:** 2021-12-04

**Authors:** Satoshi Ebata, Ayumi Yoshizaki, Takemichi Fukasawa, Asako Yoshizaki-Ogawa, Yoshihide Asano, Kosuke Kashiwabara, Koji Oba, Shinichi Sato

**Affiliations:** 1Department of Dermatology, Graduate School of Medicine, The University of Tokyo, Tokyo 113-8655, Japan; EBATAS-DER@h.u-tokyo.ac.jp (S.E.); takemichi.giraffe@gmail.com (T.F.); asako56planetes730@yahoo.co.jp (A.Y.-O.); Y_Asano_Univ@yahoo.co.jp (Y.A.); ShinSato.TokyoU@aol.com (S.S.); 2Clinical Research Support Center, The Tokyo University Hospital, Tokyo 113-8655, Japan; kashiwabara-tky@umin.ac.jp; 3Department of Biostatistics, School of Public Health, Graduate School of Medicine, and Interfaculty Initiative in Information Studies, The University of Tokyo, Tokyo 113-0033, Japan; oba@epistat.m.u-tokyo.ac.jp

**Keywords:** forced vital capacity, peripheral blood marker, red blood cell distribution width, systemic sclerosis, systemic sclerosis-associated interstitial lung disease

## Abstract

The course of systemic sclerosis-associated interstitial lung disease (SSc-ILD) varies among individuals. Red blood cell distribution width (RDW) has been reported to be a predictor of idiopathic pulmonary fibrosis. However, there are no studies on the relationship between RDW and SSc-ILD. We conducted a retrospective study of 28 patients who were diagnosed with SSc-ILD on their first visit to our hospital and were followed-up for 5 years. The correlation between the changes in RDW, KL-6, and SP-D (ΔRDW, ΔKL-6, ΔSP-D) and the changes in percent-predicted forced lung volume and % carbon monoxide diffusion (Δ%FVC, Δ%DLco) was investigated. ΔRDW at 1 year after diagnosis was significantly inversely correlated with Δ%FVC at 5 years after diagnosis (r = −0.51, *p* < 0.001) and Δ%DLco at 5 years after diagnosis (r = −0.47, *p* < 0.001), whereas ΔKL-6 and ΔSP-D at 1 year were not correlated with Δ%FVC or Δ%DLco at 5 years. In the group of SSc-ILD patients with RDW increase in the first year after diagnosis, %FVC and %DLco were significantly lower than baseline at 3-, 4-, and 5-year assessments. In the group of patients without RDW increase in the first year, %FVC and %DLco did not decrease during the follow-up period. In conclusion, the changes in RDW in the first year after diagnosis may be useful surrogate markers to predict the long-term course of SSc-ILD.

## 1. Introduction

Systemic sclerosis (SSc) is an autoimmune disease characterized by vascular damage and fibrosis in various organs such as skin and lungs [1,2,3]. SSc has a poor prognosis, especially when complicated by SSc-associated interstitial lung disease (SSc-ILD) [4,5,6]. However, the progression of SSc-ILD varies among individuals, making it difficult to determine which patients should be treated [3,7,8,9]. It would be desirable to identify simple prognostic factors in order to make better decisions about whether early treatment should be initiated in individual SSc-ILD patients to prevent future lung function deterioration.

Among the biomarkers of SSc-ILD that can be measured by blood sampling, Krebs von den Lungen (KL-6) and surfactant protein-D (SP-D) are well known. KL-6 has been suggested to be effective in predicting the response to treatment, while SP-D is effective in diagnosis and predicting the response to treatment [10]. However, the usefulness of both biomarkers as prognostic factors has not been clarified so far, and further studies are needed [10].

Coefficient of variation of red blood cell distribution width (RDW-CV) is a measure of the variability in size of red blood cell volume that is routinely reported as part of a standard complete blood count. RDW has been found to correlate with C-reactive protein and erythrocyte sedimentation rate, and is expected to be an inexpensive inflammatory marker [11]. It has also been suggested that RDW may be an inflammatory marker in patients with rheumatoid arthritis [12]. In addition, it has been proposed that RDW may be a marker of disease activity in many autoimmune diseases such as rheumatoid arthritis, systemic lupus erythematosus, and Sjogren’s syndrome [13,14,15,16,17,18]. The clinical significance of RDW in SSc is being investigated. It was reported that RDW in SSc patients correlated with inflammatory markers and that an increase in RDW over 1 year correlated with a decrease in diffusing capacity of the lung for carbon monoxide (DLco) over the same period [19]. It has also been reported that high RDW was a predictor of pulmonary hyper attention (PAH) in SSc patients [20]. Regarding lung fibrosis, there is a study suggesting that RDW is a prognostic factor for idiopathic pulmonary fibrosis [21]. However, there are no studies that have investigated the relationship between RDW and the long-term clinical course of SSc-ILD.

Herein, we examined the correlation between short-term changes in RDW and long-term changes in percent-predicted forced lung capacity (%FVC) and %DLco in SSc-ILD patients. The aim of this study was to investigate whether the changes in RDW in the early period after the diagnosis of SSc-ILD can predict future deterioration in lung function.

## 2. Materials and Methods

### 2.1. Patients

All patients who presented to our hospital (Department of Dermatology, University of Tokyo Hospital) for the first time between 2008 and 2014, had not been previously treated with immunosuppressive agents, antifibrotic agents, or biologics, and were diagnosed with SSc-ILD on initial examination at our hospital were included in this study. These patients were referred to our hospital from other hospitals and clinics because they were suspected to have SSc based on symptoms such as skin sclerosis and Raynaud’s phenomenon. Patients who missed annual pulmonary function tests or those who developed PAH within 5 years were excluded. The diagnosis of SSc was made according to the 2013 American College of Rheumatology/European League against Rheumatism classification criteria [22]. The presence of SSc-ILD was determined by radiologists based on high-resolution computed tomography images. This study was approved by the ethics committee at The University of Tokyo Hospital and was conducted in accordance with the principles of the Declaration of Helsinki. Written informed consent was obtained from all patients.

### 2.2. Data Collection

We retrospectively reviewed the electronic medical records. For patients’ demographics, data were recorded at the time of their first visit. Disease duration was defined as time from onset of the first non-Raynaud’s phenomenon manifestation. Skin involvement was scored according to the modified Rodnan skin score [23]. Patients were divided into diffuse and limited cutaneous subsets by the criteria of LeRoy [1]. Laboratory data, such as RDW, were collected at the time of the first visit and one year later. Respiratory function tests were performed in accordance with international guidelines [24]. %FVC and %DLco data were collected at the time of the first visit and at one, two, three, four, and five years after that.

### 2.3. Definition of Progressive Fibrosing ILD

Several previous clinical trials have defined progressive fibrosing ILD as a decrease in FVC of 10% or more, or a decrease in FVC of 5–9% combined with a decrease in DLco of 15% or more [25,26]. The same criteria were adopted in the present study, and patients who met the criteria from the time of initial diagnosis to 5 years later were considered to have progressive fibrosing ILD.

### 2.4. Statistical Analysis

All the data were statistically analyzed using GraphPad Prism version 8.0 statistical software (GraphPad Software Inc., San Diego, CA, USA). Quantitative data were presented as mean ± standard deviation [SD]. Significance tests for comparisons between groups were based on Wilcoxson test and Mann–Whitney U-test. Differences were considered statistically significant at *p* < 0.05.

## 3. Results

### 3.1. Patients’ Demographics

Twenty-eight patients were included. Five patients who missed follow-up pulmonary function tests and one patient who developed PAH within 5 years of their initial diagnosis were excluded from the current analysis. Patients’ demographics are shown in Table 1. The baseline %FVC was 81.2 ± 2.3%, and %DLco was 82.0 ± 2.9%. There were 13 patients (46%) whose RDW changes (ΔRDW) in the first year after initial examination at our clinic was 0 or less, and 15 patients (54%) whose ΔRDW in 1 yr. was greater than 0. There were no statistically significant differences in baseline items between the group of patients with RDW increase in the first year and the group of patients without RDW increase in the first year. The baseline mRSS and disease duration of the group with increased RDW in the first year were 16.6 ± 1.7 and 24.6 ± 3.9 months, respectively, whereas the baseline mRSS and disease duration of the group with no increase in RDW in the first year were 17.2 ± 2.6 and 24.8 ± 5.1 months, respectively. In both groups, the baseline mRSS and disease duration were almost the same and unrelated to the subsequent increase or decrease in RDW.

### 3.2. The Correlation between Short-Term ΔRDW and Long-Term Changes in Pulmonary Function

We examined the correlation between the changes in RDW, KL-6 (ΔKL-6), and SP-D (ΔSP-D) in 1 year after first admission and those in %FVC (ΔFVC) and %DLco (ΔDLco) in 5 years. Correlation analyses showed that ΔRDW in 1 year negatively correlated with both Δ%FVC in 5 years (r = −0.51, *p* < 0.001; Figure 1A) and Δ%DLco in 5 years (r = −0.47, *p* < 0.001; Figure 1D). ΔKL-6 in 1 year and ΔSP-D in 1 year did not correlate with Δ%FVC in 5 years (r = −0.04, *p* = 0.31; Figure 1B) (r = −0.03, *p* = 0.42; Figure 1E) or Δ%DLco in 5 years (r = −0.02, *p* = 0.44; Figure 1C) (r = 0.003, *p* = 0.77; Figure 1F).

### 3.3. Clinical Course of SSc-ILD Stratified by ΔRDW

In the subgroup of patients without RDW increase in the first year, there was no decrease in %FVC or %DLco throughout 5 years (Figure 2A,B). Δ%FVC in 5 years was 4.85 ± 2.10% and Δ%DLco in 5 years was 5.09 ± 2.85%. In another subgroup of patients with RDW increase in the first year, %FVC was significantly worse than baseline at 3, 4, and 5 years (*p* = 0.008, <0.001, <0.001, respectively; Figure 2A). %DLco was also significantly lower than baseline at 3, 4, and 5 years (*p* < 0.001, <0.001, <0.001, respectively; Figure 2B). Δ%FVC in 5 years was −12.80 ± 3.11% and Δ%DLco in 5 years was −11.30 ± 4.17%.

The number of patients who were treated with immunosuppressants during 5 years was five (38.5%) in the subgroup of patients without RDW increase in the first year, and nine (60.0%) in another subgroup of patients with RDW increase in the first year. In an analysis of only patients who did not receive immunosuppressive drugs in 5 years, patients without RDW increase in the first year had a ΔFVC of 2.75 ± 8.22% and a ΔDLco of 7.12 ± 13.8% in the fifth-year evaluation. In contrast, patients with RDW increase in the first year had a ΔFVC of −11.6 ± 10.9% and a ΔDLco of −10.9 ± 12.2% at year 5 (Figure 2C,D). In the group of patients with RDW increase in the first year, %FVC at 5 years was significantly lower than baseline (*p* = 0.03; Figure 2C), and the %DLco at 5 years showed a downward trend compared to baseline (*p* = 0.05; Figure 2D). None of patients received antifibrotic drugs or biologics during the follow-up period.

### 3.4. Sensitivity and Specificity of Identifying Progressive Fibrosing IPF by RDW Elevation

In 13 patients whose ΔRDW in 1 year was 0 or less, there were no cases of progressive fibrosing ILD. In 15 patients whose ΔRDW in 1 year was greater than 0, there were 8 patients with progressive fibrosing ILD, and they all had a >10% decrease in %FVC over 5 years. In other words, ΔRDW in 1 year greater than 0 had a sensitivity of 100% and a specificity of 65% for identifying progressive fibrosing ILD at 5 years.

## 4. Discussion

By this retrospective study, we demonstrated that the increase in RDW, a quantitative measure of variability in the size of circulating erythrocytes, at early stage of SSc-ILD may predict the long-term deterioration of %FVC and %DLco. This is the first study to show that RDW may be a useful prognostic predictor of SSc-ILD.

First, we revealed that ΔRDW in 1 year after diagnosis of SSc-ILD was inversely correlated with both Δ%FVC and Δ%DLco in 5 years, while ΔKL-6 and ΔSP-D did not correlate with Δ%FVC or Δ%DLco. KL-6 and SP-D, which reflect alveolar epithelial cell damage and dysfunction, are most popular serum markers of SSc-ILD [10].

Second, throughout the 5 years after diagnosis, %FVC and %DLco were higher than baseline in the subgroup of patients without RDW increase in the first year, and lower than baseline in another subgroup of patients with RDW increase in the first year. These trends were also observed in a sub-analysis of patients who did not receive immunosuppressive therapy. If ΔRDW in 1 year was less than or equal to 0, no patient had a progressive fibrosing ILD at 5 years.

Based on these results, RDW may be a biomarker that can be measured in blood samples to predict the long-term prognosis of SSc-ILD. It is currently difficult to predict the course of SSc-ILD at an early stage. In recent years, several drugs have been shown to be effective in the treatment of SSc-ILD [27,28,29], but it is not clear in which groups of patients these drugs should be considered. In addition, it is not advisable to administer these drugs to patients with SSc-ILD who are not expected to progress because of their disadvantages such as high side effects and high cost. Our data suggest that if the ΔRDW is 0 or less, the risk of developing progressive fibrosing ILD is low, and the decision to introduce therapeutic agents in these patients may be made with caution. However, how to combine and use RDW with KL-6, SP-D and other biomarkers to find patients who need to be introduced to therapeutic agents will need to be further investigated.

The limitation of this study is that it was a retrospective study with a small number of subjects. Although comorbidities such as anemia, iron deficiency, folate deficiency, and vitamin B12 deficiency may affect elevated RDW, these confounders could not be excluded from the analysis due to the limited number of cases in this study. Even though there is no difference in baseline characteristics between the groups with increased RDW and those without increased RDW, it would be desirable to perform a multivariate linear regression analysis with the addition of variables such as baseline mRSS and disease duration to better clarify the impact of RDW. In the future, multivariate linear regression analysis using larger prospective data is needed to completely eliminate the influence of confounding factors and to elucidate the clinical significance of RDW in predicting the course of SSc-ILD.

The strength is that RDW is a simple and inexpensive indicator reported in a standard blood test. It will be clinically significant if monitoring RDW helps to select the best treatment for individual SSc-ILD patients.

In conclusion, we first suggested that short-term changes in RDW can be helpful in predicting the course of SSc-ILD. Monitoring of RDW may be important in the follow-up of patients with early SSc-ILD.

## Figures and Tables

**Figure 1 diagnostics-11-02274-f001:**
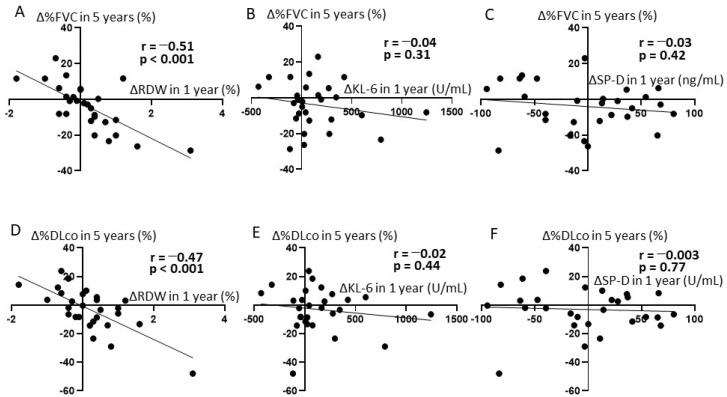
Correlation of peripheral blood markers with the absolute changes in %FVC and %DLco in SSc-ILD patients. (**A**) The absolute change of RDW at year 1 vs. absolute change in %FVC at year 5. (**B**) The absolute change of KL-6 at year 1 vs. absolute change in %FVC at year 5. (**C**) The absolute change of SP-D at year 1 vs. absolute change in %FVC at year 5. (**D**) The absolute change of RDW at year 1 vs. absolute change in %DLco at year 5. (**E**) The absolute change of KL-6 at year 1 vs. absolute change in %DLco at year 5. (**F**) The absolute change of SP-D at year 1 vs. absolute change in %DLco at year 5.

**Figure 2 diagnostics-11-02274-f002:**
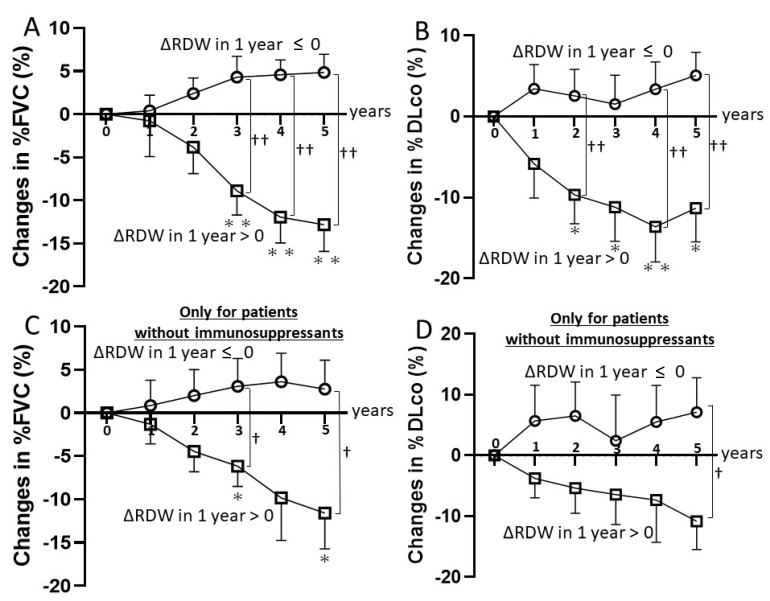
Comparison of SSc-ILD progression stratified by the changes in RDW. (**A**) Changes in %FVC in the group of patients with RDW changes in the first year greater than 0 and in the group of patients with RDW changes in the first year 0 or less. (**B**) Changes in %DLco in the group of patients with RDW changes in the first year greater than 0 and in the group of patients with RDW changes in the first year 0 or less. (**C**) Changes in %FVC in the group of patients with RDW changes in the first year greater than 0 and in the group of patients with RDW changes in the first year 0 or less among patients without immunosuppressants for SSc-ILD in 5 years. (**D**) Changes in %DLco in the group of patients with RDW changes in the first year greater than 0 and in the group of patients with RDW changes in the first year 0 or less among patients without immu-nosuppressants for SSc-ILD in 5 years. The absolute changes between the two groups were compared at 1-, 2-, 3-, 4-, and 5-year assessments. All values represent the mean ± SD. * *p* < 0.05 and ** *p* < 0.01 vs. baseline. † *p* < 0.05 and †† *p* < 0.01 vs. the group whose changes of RDW in 1 year were greater than 0.

**Table 1 diagnostics-11-02274-t001:** Baseline characteristics of patients.

Characteristic	All (*n* = 28)	ΔRDW ≤ 0 (*n* = 13)	ΔRDW > 0 (*n* = 15)
Female sex, no. (%)	25 (89.3)	11 (84.6)	14 (93.3)
Age, years	53.1 ± 2.5	50.7 ± 2.8	55.2 ± 4.0
Diffuse cutaneous systemic sclerosis, no. (%)	22 (78.6)	10 (76.9)	12 (80.0)
Disease duration, months	24.6 ± 3.9	24.5 ± 6.0	24.8 ± 5.1
Range	3–71	3–71	5–69
Modified Rodnan skin score	16.6 ± 1.7	17.2 ± 2.6	16.1 ± 2.2
Smoking history, % yes	17.9%	15.4%	20.0%
RDW, %	13.6 ± 0.1	13.7 ± 0.2	13.5 ± 0.2
FVC, % of predicted value	81.2 ± 2.3	80.1 ± 3.6	82.1 ± 3.3
DL_CO_, % of predicted value	82.0 ± 2.9	80.0 ± 4.9	84.0 ± 4.0
KL-6, U/mL	765.4 ± 98.0	793.1 ± 144.0	673.3 ± 135.2
SP-D, ng/mL	111.9 ± 12.8	117.0 ± 22.9	101.3 ± 15.1
Hemoglobin, g/dL	12.7 ± 0.2	12.7 ± 0.3	12.9 ± 0.3
Anti-topoisomerase I antibody positive, no. (%)	20 (71.4)	9 (69.2)	11 (73.3)

Unless otherwise indicated, values are means ± standard deviation. All the clinical and laboratory parameters were obtained at the first evaluation. DL_CO_, diffusion capacity for carbon monoxide; FVC, forced vital capacity; KL-6, Krebs von den Lungen-6; RDW, red blood cell distribution width; SP-D, surfactant protein-D.

## Data Availability

Anonymized data available upon request.

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
