# Peer review of "Increased Red Blood Cell Distribution Width in the First Year after Diagnosis Predicts Worsening of Systemic Sclerosis-Associated Interstitial Lung Disease at 5 Years: A Pilot Study"

_diagnostics, 2021, doi:10.3390/diagnostics11122274_

Round 1

Reviewer 1 Report

Ebata et al. present a retrospective cohort study in systemic sclerosis associated interstitial lung disease (SSc-ILD) patients examining the predictive value of increased RDW at one year after diagnosis correlating to outcomes of FVC, DLco, KL-6, and SP-D.  The authors find an increase in RDW greater than 0 at year 1 correlates to a decreased FVC and DLco in patients in patients at 3, 4, and 5 years after diagnosis.  While this correlation between RDW and lung function has been shown as a prognostic indicator in idiopathic pulmonary fibrosis, expanding understanding to (SSc-ILD) is a significant finding.

Overall, this is interesting report that provides important findings that can help guide clinical decision making in the treatment of early SSc-ILD.  The work is logical, although there are a number of issues that the authors need to address to support the impact of their work.

Major comments:

  1. Methods. Can the authors provide more detail on the clinic(s) at which the data was collected as well as how patients were referred to the clinic?

  1. Methods. How many patients were excluded due to each missed pulmonary function testing and pulmonary arterial hypertension?

  1. In the figure legends the reader should be able to make their own conclusions from data and then go to the results section to see the authors’ conclusions. There are too many conclusions in the figure legends currently. For example, Figure 1 legend “(A), The absolute change of RDW in 1 year after diagnosis negatively correlate with the absolute change of %FVC in 5 years.” should be similar to “(A), The absolute change of RDW at year 1 vs absolute change in %FVC at year 5.”

  1. To provide the clinician reading more information on how the results should impact their clinical process there should be an attempt to provide data on sensitivity or specificity for increased RDW to decline in FVC or DLco.

  1. Figure 2. It would be helpful to add labels to show that A-B are examining all patients and C-D are examining patients who did not receive immunosuppression.

  1. Question. Did the authors examine how the RDW correlated in patients who did receive immunosuppression?

Minor comments:

  1. Line 68. Error in citation placement “LeRoy.1” to “LeRoy [1].”

Author Response

Comments to be transmitted to the Author

 Ebata et al. present a retrospective cohort study in systemic sclerosis associated interstitial lung disease (SSc-ILD) patients examining the predictive value of increased RDW at one year after diagnosis correlating to outcomes of FVC, DLco, KL-6, and SP-D.  The authors find an increase in RDW greater than 0 at year 1 correlates to a decreased FVC and DLco in patients in patients at 3, 4, and 5 years after diagnosis.  While this correlation between RDW and lung function has been shown as a prognostic indicator in idiopathic pulmonary fibrosis, expanding understanding to (SSc-ILD) is a significant finding.

Overall, this is interesting report that provides important findings that can help guide clinical decision making in the treatment of early SSc-ILD.  The work is logical, although there are a number of issues that the authors need to address to support the impact of their work.

Response

We would like to express our deepest gratitude to Reviewer 1 for pointing out the important issues that we need to address and for his valuable suggestions on how to solve them. Below, we would like to respond to Reviewer 1's points one by one.

Comment 1

Major comments:

Methods. Can the authors provide more detail on the clinic(s) at which the data was collected as well as how patients were referred to the clinic?

Response

Thank you for your valuable suggestion. We apologize for the lack of explanation. We added the following on page 2, line 66, and would appreciate it if you could check it.

“All patients who presented to our hospital (Department of Dermatology, University of Tokyo Hospital) for the first time between 2008 and 2014, had not been previously treated with immunosuppressive agents, antifibrotic agents, or biologics, and were diagnosed with SSc-ILD on initial examination at our hospital were included in this study. These patients were referred to our hospital from other hospitals and clinics because they were suspected to have SSc based on symptoms such as skin sclerosis and Raynaud's phenomenon.”

Comment 2

Methods. How many patients were excluded due to each missed pulmonary function testing and pulmonary arterial hypertension?

Response

 Thank you for pointing this out. We have added the following details about the excluded person in line 108 on page 3. We appreciate you taking the time to check.

 “Five patients who missed follow-up respiratory function tests and one patient who developed pulmonary arterial hypertension within 5 years of the initial diagnosis were excluded from the current analysis.”

Comment 3

 In the figure legends the reader should be able to make their own conclusions from data and then go to the results section to see the authors’ conclusions. There are too many conclusions in the figure legends currently. For example, Figure 1 legend “(A), The absolute change of RDW in 1 year after diagnosis negatively correlate with the absolute change of %FVC in 5 years.” should be similar to “(A), The absolute change of RDW at year 1 vs absolute change in %FVC at year 5.”

Response

Thank you for your kind guidance. We are very sorry for the inappropriate description. We have changed the figure legends and highlighted it in yellow, so we hope you will check it.

Comment 4

To provide the clinician reading more information on how the results should impact their clinical process there should be an attempt to provide data on sensitivity or specificity for increased RDW to decline in FVC or DLco.

Response

 Thank you very much for your valuable feedback. Together with the suggestion of reviewer 2, we have addressed this issue. We defined progressive fibrosing ILD and determined the sensitivity and specificity. Specifically, the following additions were made to the "Methods," "Results," and "Discussion" sections, respectively.

Method

“2.3. Definition of progressive fibrosing ILD

Several previous clinical trials have defined progressive fibrosing ILD as a decrease in FVC of 10% or more, or a decrease in FVC of 5-9% combined with a decrease in DLco of 15% or more [25,26]. The same criteria were adopted in the present study, and patients who met the criteria from the time of initial diagnosis to 5 years later were considered to have progressive fibrosing ILD.”

Results

“3.4. Sensitivity and specificity of identifying progressive fibrosing IPF by RDW elevation

In 13 patients whose ΔRDW in 1yr. was 0 or less, there were no cases of progressive fibrosing ILD. In 15 patients whose ΔRDW in 1yr. was greater than 0, there were 8 patients with progressive fibrosing ILD, and they all had a >10% decrease in %FVC over 5 years. In other words, ΔRDW in 1yr. greater than 0 had a sensitivity of 100% and a specificity of 65% for identifying progressive fibrosing ILD at 5 years.”

Discussion

“If ΔRDW in 1 yr. was less than or equal to 0, no patient had a progressive fibrosing ILD at 5 years.”

“Our data suggest that if the ΔRDW is 0 or less, the risk of developing progressive fibrotic ILD is low, and the decision to introduce therapeutic agents in these patients may be made with caution.”

Comment 5

Figure 2. It would be helpful to add labels to show that A-B are examining all patients and C-D are examining patients who did not receive immunosuppression.

Response

 Thank you for your appropriate suggestions. We added the labels in Figure 2 and would appreciate if you could confirm them.

Comment 6

 Question. Did the authors examine how the RDW correlated in patients who did receive immunosuppression?

Response

 We decided that the analysis of only patients who received immunosuppressive therapy was not worth including in the manuscript because the number of patients was too small to make a judgment based on this alone, and the results were almost identical to the overall patient group. But if there is a need for additional information, please let us know.

Comment 7

Minor comments:

Line 68. Error in citation placement “LeRoy.1” to “LeRoy [1].”

Response

Thank you for the instruction. We are sorry for the mistake. We corrected it, so please confirm.

Reviewer 2 Report

The quest for a reliable and simple biomarker to predict long-term progression of SSc-ILD is a clinically meaningful area of research. However, I feel that this paper, especially due to its very small sample size, does not help to expand this subject area.

In my opinion, the introduction does not provide adequate context for the present work. Regarding the SSc-ILD progression are given only very general information. There is no mention of previous researches regarding biomarkers which may predict lung fibrosis worsening. Two of these previously studied biomarkers – KL-6 and SP-D – are mentioned in the abstract and appeared (quite suddenly) in the results section. However, their value and limitations, interpretation of previous studies are not discussed.  Similarly, the value of RDW as a biomarker is superficially addressed. To say that “RDW reflects inflammation” oversimplify a whole body of research in various diseases. Only the link with idiopathic pulmonary fibrosis is mentioned, and not correctly referenced (e.g the ref 10 is a retrospective analysis of healthy blood donors, in which was proven an association of RDW with age). Other studies in the recent literature addressed this subject – utility of RDW as a biomarker in various immune diseases, including systemic sclerosis – however, these were not mentioned. Also, there is no mention of many potential confounders for RDW increasing including various types of anemia, nutritional deficiencies (folate, vit B12, etc), various chronic diseases, etc. In addition, these issues were not addressed by clear exclusion criteria, neither by mentioning them at the study limitation section. Also, no statistical methods were used to control for confounding factors, probably due to very small sample size.

I think that discussion should take into account in more depth sources of potential sources of bias and imprecision of these biomarkers. Moreover, a discussion about what it is considered a clinically meaningful progression of SSc-ILD (there are published guidelines on this topic) is probably needed, if relation with treatment strategy is mentioned based on these results.

The conclusions seems to be overstated, as no definite conclusion or superiority of RDW to other potential biomarkers, as KL-6 and SP-D, can be made from a retrospective small observational study. Even more so, based on these results one cannot conclude that RDW can be a biomarker for treatment initiation and/or treatment selection. In my opinion, is needed a more cautious overall interpretation of these results, considering the study limitations, and results from similar studies.

Author Response

Comments for the Author

The quest for a reliable and simple biomarker to predict long-term progression of SSc-ILD is a clinically meaningful area of research. However, I feel that this paper, especially due to its very small sample size, does not help to expand this subject area.

In my opinion, the introduction does not provide adequate context for the present work.

Response

Thank you for your valuable suggestions. We are very sorry for the many inadequacies. As you pointed out, this study is preliminary. We believe that the accumulation of large scale data is necessary in the future. In order to clarify this point, we added the phrase "A Pilot Study" to the title. In addition, we responded to each comment as follows.

Comment 1

 Regarding the SSc-ILD progression are given only very general information. There is no mention of previous researches regarding biomarkers which may predict lung fibrosis worsening. Two of these previously studied biomarkers – KL-6 and SP-D – are mentioned in the abstract and appeared (quite suddenly) in the results section. However, their value and limitations, interpretation of previous studies are not discussed.

Response

 We cited a review article on SSc-ILD biomarkers and added the following on page 1, line 38, regarding the clinical significance of both as currently known.

 “Among the biomarkers of SSc-ILD that can be measured by blood sampling, Krebs von den Lungen (KL-6) and surfactant protein‐D (SP-D) are well known. KL-6 has been suggested to be effective in predicting the response to treatment, while SP-D is effective in diagnosis and predicting the response to treatment [10]. However, the usefulness of both biomarkers as prognostic factors has not been clarified so far, and further studies are needed [10].”

Comment 2

 Similarly, the value of RDW as a biomarker is superficially addressed. To say that “RDW reflects inflammation” oversimplify a whole body of research in various diseases. Only the link with idiopathic pulmonary fibrosis is mentioned, and not correctly referenced (e.g the ref 10 is a retrospective analysis of healthy blood donors, in which was proven an association of RDW with age). Other studies in the recent literature addressed this subject – utility of RDW as a biomarker in various immune diseases, including systemic sclerosis – however, these were not mentioned.

Response

 Thank you for pointing this out. We apologize for the inadequate and inaccurate description. We have changed the description as follows on page 2, line 46, based on appropriate reference materials.

“RDW has been found to correlate with C-reactive protein and erythrocyte sedimentation rate, and is expected to be an inexpensive inflammatory marker [11]. It has also been suggested that RDW may be an inflammatory marker in patients with rheumatoid arthritis [12]. In addition, it has been proposed that RDW may be a marker of disease activity in many autoimmune diseases such as rheumatoid arthritis, systemic lupus erythematosus, and Sjogren's syndrome [13-18]. The clinical significance of RDW in SSc is being investigated. It was reported that RDW in SSc patients correlated with inflammatory markers and that an increase in RDW over 1 year correlated with a decrease in diffusing capacity of the lung for carbon monoxide (DLco) over the same period [19]. It has also been reported that high RDW was a predictor of pulmonary hyper attention (PAH) in SSc patients [20]. Regarding lung fibrosis, there is a study suggesting that RDW is a prognostic factor for idiopathic pulmonary fibrosis [21].”

Comment 3

Also, there is no mention of many potential confounders for RDW increasing including various types of anemia, nutritional deficiencies (folate, vit B12, etc), various chronic diseases, etc. In addition, these issues were not addressed by clear exclusion criteria, neither by mentioning them at the study limitation section. Also, no statistical methods were used to control for confounding factors, probably due to very small sample size. I think that discussion should take into account in more depth sources of potential sources of bias and imprecision of these biomarkers.

Response

 Thank you for your valuable suggestion. We agree with your concerns. We have added the following statement to the limitation on page 7, line 226. In addition, we have added the words "a pilot study" to the title to clarify that this study is still preliminary.

 “Although comorbidities such as anemia, iron deficiency, folate deficiency, and vitamin B12 deficiency may affect the elevated RDW, these confounders could not be excluded from the analysis due to the limited number of cases in this study. Future studies with larger prospective data are needed to clarify the clinical significance of RDW in predicting SSC-ILD without the influence of confounders.”

Comment 4

Moreover, a discussion about what it is considered a clinically meaningful progression of SSc-ILD (there are published guidelines on this topic) is probably needed, if relation with treatment strategy is mentioned based on these results.

Response

  Thank you very much for your valuable feedback. Together with the suggestion of reviewer 1, we have addressed this issue. We defined progressive fibrosing ILD and determined the sensitivity and specificity. Specifically, the following additions were made to the "Methods," "Results," and "Discussion" sections, respectively.

Method

“2.3. Definition of progressive fibrosing ILD

Several previous clinical trials have defined progressive fibrosing ILD as a decrease in FVC of 10% or more, or a decrease in FVC of 5-9% combined with a decrease in DLco of 15% or more [25,26]. The same criteria were adopted in the present study, and patients who met the criteria from the time of initial diagnosis to 5 years later were considered to have progressive fibrosing ILD.”

Results

“3.4. Sensitivity and specificity of identifying progressive fibrosing IPF by RDW elevation

In 13 patients whose ΔRDW in 1yr. was 0 or less, there were no cases of progressive fibrosing ILD. In 15 patients whose ΔRDW in 1yr. was greater than 0, there were 8 patients with progressive fibrosing ILD, and they all had a >10% decrease in %FVC over 5 years. In other words, ΔRDW in 1yr. greater than 0 had a sensitivity of 100% and a specificity of 65% for identifying progressive fibrosing ILD at 5 years.”

Discussion

“If ΔRDW in 1 yr. was less than or equal to 0, no patient had a progressive fibrosing ILD at 5 years.”

“Our data suggest that if the ΔRDW is 0 or less, the risk of developing progressive fibrotic ILD is low, and the decision to introduce therapeutic agents in these patients may be made with caution.”

Comment 5

 The conclusions seems to be overstated, as no definite conclusion or superiority of RDW to other potential biomarkers, as KL-6 and SP-D, can be made from a retrospective small observational study. Even more so, based on these results one cannot conclude that RDW can be a biomarker for treatment initiation and/or treatment selection. In my opinion, is needed a more cautious overall interpretation of these results, considering the study limitations, and results from similar studies.

Response

 Thank you for your guidance. We agree with your concerns. In order to avoid misunderstanding, we removed potentially exaggerated expressions such as the comparison between RDW and KL-6 or SP-D, and replaced them with the following on page 6, line 214. Please also review my reply to comment 4 above regarding treatment strategies.

 “Based on these results, RDW may be a biomarker that can be measured in blood samples to predict the long-term prognosis of SSc-ILD. It is currently difficult to predict the course of SSc-ILD at an early stage. In recent years, several drugs have been shown to be effective in the treatment of SSc-ILD [27-29], but it is not clear in which groups of patients these drugs should be considered. In addition, it is not advisable to administer these drugs to patients with SSc-ILD who are not expected to progress because of their disadvantages such as high side effects and high cost. Our data suggest that if the ΔRDW is 0 or less, the risk of developing progressive fibrosing ILD is low, and the decision to introduce therapeutic agents in these patients may be made with caution. However, how to combine and use RDW with KL-6, SP-D and other biomarkers to find patients who need to be introduced to therapeutic agents will need to be further investigated.”

Round 2

Reviewer 2 Report

Thank you for editing your paper based on all concerns I have initially expressed. In my opinion, the clinical significance of your research will be much clear for your readers. I recommend publication of the manuscript.  

Author Response

We appreciate your very kind comments. Thanks to your earlier suggestion, we were able to improve the content of the paper. Please accept my sincere thanks.